# The effect of ice supersaturation and thin cirrus on lapse rates in the upper troposphere

Klaus Gierens[1], Lena Wilhelm[1,3], Sina Hofer[1], and Susanne Rohs[2]

[1]Deutsches Zentrum für Luft- und Raumfahrt, Institut für Physik der Atmosphäre, Oberpfaffenhofen, Germany
[2]Forschungszentrum Jülich, IEK-8, Jülich, Germany
[3]now at: Oeschger Centre for Climate Change Research and Institute of Geography, University of Bern, Bern, Switzerland

**Correspondence:** Klaus Gierens (klaus.gierens@dlr.de)

**Abstract.** In this paper, the effects of ice supersaturated regions and thin, subvisual cirrus clouds on lapse rates are examined. For that, probability distribution and density functions of the lapse rate and the potential temperature gradients from ten years of measurement data from the MOZAIC/IAGOS project and ERA-5 reanalysis data were produced, and an analysis of an example case of an ice supersaturated region with a large vertical extent is performed. For the study of the probability distribution and density functions, a distinction is made between ice subsaturated, ice supersaturated air masses (persistent contrails) and situations of particularly high ice supersaturation that allow the formation of optically thick and strongly warming contrails. The estimation of the lapse rates involves two adjacent standard pressure levels of the reanalysis surrounding MOZAIC´s measurement/flight points. If the upper of these levels is in the stratosphere, the distribution function for subsaturated cases shows much lower lapse rates than those of supersaturated cases. If all levels are in the troposphere the distributions become more similar, but the average lapse rates are still higher in supersaturated than in subsaturated cases and the distributions peak at higher values and are narrower in ISSRs than elsewhere. This narrowing is particularly pronounced if there is substantial supersaturation.

For the examination of an example case, ERA-5 data and forecasts from ICON-EU (DWD) are compared. ERA-5 data, in particular, shows a large ice supersaturated region below the tropopause, that was pushed up by uplifting air, while the data of ICON-EU indicates areas of saturation. The lapse rate in this ice supersaturated region (ISSR), which is large, is associated with clouds and high relative humidity. Supersaturation and cloud formation result from uplifting of air layers. The temperature gradient within an uplifting layer steepens, both for dry and moist air. As soon as condensation or ice formation starts in the upper part of a lifting layer, the release of latent heat begins to decrease the lapse rate, but radiation starts to act in the opposite direction, keeping the lapse rate high. The highest lapse rates close to the stability limit can only be reached in potentially unstable situations.

## 1 Introduction

Although the existence of ice supersaturation in the atmosphere is known for more than 100 years (Wegener, 1914; Gierens et al., 2012), the investigation of physical and dynamical characteristics of ice supersaturated regions (ISSRs) only commenced about 25 years ago, mainly because the then-new MOZAIC project (Measurement of OZone and water vapor by Airbus

In-service airCraft, Marenco et al., 1998) started delivering large amounts of humidity data in the upper troposphere where ISSRs are quite frequent. The airborne data have been used to demonstrate that the degree of supersaturation within ISSRs is exponentially distributed (Gierens et al., 1999) and how long aircraft fly on average through ISSRs during their cruise (Gierens and Spichtinger, 2000). A typical path length is 150 km, but the distribution is heavily skewed to larger values. The MOZAIC data showed that air within midlatitude ISSRs is generally a bit colder and a bit moister than air in the subsaturated neighbourhood. The additional use of satellite data corroborated these results, extended them to ISSRs in the tropics and even over Antarctica (Spichtinger et al., 2003b), and demonstrated a close relation of ice supersaturation to sub-visible cirrus (Gierens et al., 2000). Using high-resolution radiosonde data for which correction algorithms were applied to the raw humidity data made it possible to derive a thickness distribution for ISSRs (Spichtinger et al., 2003a) with a typical value of a few hundred metres, again with a skewed distribution. Later studies using airborne, satellite, or corrected radiosonde data refined but essentially confirmed these early results (for a review, see Gierens et al., 2012, and references therein).

The European Centre for Medium-Range Weather Forecasts (ECMWF) was the first weather centre to introduce ice supersaturation in their forecast model in 2006 (model version C31R1 of the integrated forecast system, IFS, Tompkins et al., 2007) and consequently, ice supersaturation was later also represented in its reanalysis products ERA-Interim (Dee et al., 2011) and the most recent one ERA-5 (Hersbach et al., 2018a, b, 2020). These data made it possible to investigate the distribution of vertical velocity, divergence and relative vorticity inside and outside of ISSRs (Gierens and Brinkop, 2012). Characteristic differences were found in a study covering Europe and the Tropics and four months of data (June, September and December 2011 and March 2021): Ice supersaturation occurs predominantly in uplifting divergent airmasses and in anticyclonic circulation.[1]

In the present paper we show that also another dynamical characteristic of airmasses, the vertical temperature gradient or lapse rate, displays substantially differing distributions in ice supersaturated and subsaturated air. We determine the lapse rate from the reanalysis' pressure and temperature values. It turns out that the lapse rate is a quantity with distinct probability distributions: ISSRs are concentrated at near-neutral conditions (high lapse rate); low lapse rates are very rare for ISSRs but not for subsaturated air.

The paper proceeds with the presentation of the data and methods and then shows two types of results, statistics from 10 years of data and as an example a case with a vertically quite extended ISSR in the upper troposphere. For this case, we use data from two weather models and thus provide *en passant* a comparison of these two. In the discussion section, we try to give explanations for the surprising coupling between thermodynamics (ISSR) and dynamics (stability). Finally, we summarize the results.

---

[1]We may occasionally use the short expression "supersaturated airmass" instead of "airmass including supersaturated water vapour", although this might be misleading. It is only the water vapour that is supersaturated. The air is not.

## 2 Data and methods

### 2.1 Data from commercial aircraft

For the first part of the study, a combination of MOZAIC In-service Aircraft for a Global Observing System (IAGOS) and ERA-5 reanalysis data was used. We could have performed the study with reanalysis data alone, like in an earlier study (Gierens and Brinkop, 2012), using temperature and humidity from the reanalysis. But we preferred to take these data from in-situ measurements, as another study (Gierens et al., 2020) has shown that the ability of ERA-5 to predict ice supersaturation at the right place and time and at the correct value is in need of improvement. The MOZAIC campaign (Marenco et al., 1998), which in 2011 was transferred into IAGOS (Petzold et al., 2015), is a European Research Infrastructure that offers global in-situ data for meteorological quantities and atmospheric chemical composition. They are produced from fully-automated equipment installed on commercial passenger aircraft. Over 62,000 flights can be accessed at the IAGOS portal (IAGOS Data Portal, 2019). For this study, only flights from MOZAIC (IAGOS CORE) of the years 2000–2009 were selected, amounting to 16,588 flights. From those flights, air temperature $T$ and relative humidity $RH$, as well as the flight position were used for the analysis. Data points were restricted to pressure altitudes at cruise level between 310 hPa and 190 hPa (9–12 km) and to the research area extending from 30°N to 70°N latitude (zone with most air traffic) and 125°W to 145°E longitude. This made sure that most of MOZAIC's flights were covered, while not including too many different climate zones. Only calibrated, reliable data, were considered. Measurements are available every 4 seconds. Temperature data have a precision of $\pm 0.2$ K and an accuracy of $\pm 0.5$ K. Relative humidity is given with a precision of 1% and an accuracy of 5% (Neis et al., 2015). Measurements with $RH > 100\%$ (RH with respect to liquid water) have been rejected, as this would indicate flying through liquid water clouds, which do not exist at temperatures where contrail formation is possible. The humidity sensor's response slows down with decreasing temperature, from 1 s at 300 K to 120 s at 215 K (Neis et al., 2015). This implies strong inertia at the cruise level of the aircraft. Fortunately, this is not a problem for the present study, since we randomly selected only one percent of the data points, such that, on a temporal scale, two consecutive data points analyzed were on average 400 s apart, much more than the mentioned 120 s inertia time scale. The selection is done using a (0-1)-uniform random number generator that selects data points only if the variate is smaller than 0.01. This ensured the independence of individual data points and excluded autocorrelation in the dataset. For each selected data point the corresponding weather data from the ERA-5 reanalysis is then collected.

### 2.2 Reanalysis data

The ERA-5 Reanalysis (Hersbach et al., 2018a, b, 2020), as the fifth generation ECMWF reanalysis for the global climate and weather, provides global and consistent analyses and forecast fields from 1950 until present, including a large number of atmospheric, land-surface, and ocean-wave variables, that are produced using 4D-Var data assimilation. ERA-5 forecasts are produced twice daily (00, 12 UTC) using ECMWF's Integrated Forecasting System (IFS). For the statistics of the lapse rate hourly ERA-5 reanalysis data were retrieved from the Copernicus Data Service (Copernicus Climate Change Service (C3S), 2017) for the same years from 2000 to 2009. Specifically, temperature data on pressure levels 200, 225, 250, and 300 hPa, are

used in $1° \times 1°$ spatial resolution for calculation of the lapse rate surrounding each of MOZAIC´s flight points. Temperature data from ERA–5 agree well with MOZAIC in-situ data and a recent comparison with radiosonde data (Bland et al., 2021) confirms the good quality of the reanalysis' temperature in the upper troposphere and lower stratosphere.

Additionally, we retrieve radiation quantities and cirrus ice water concentration to calculate the instantaneous radiative forcing (iRF) of a contrail that may have formed by the corresponding MOZAIC aircraft (as in Wilhelm et al., 2021, in the following abbreviated as WGR21). iRF is the difference between the top-of-atmosphere net radiation fluxes in a situation with a contrail and the same situation without. For this, we use the parameterization by Schumann et al. (2012). Using iRF, it is possible to distinguish supersaturated cases where contrails are persistent and have quite strong iRF from other supersaturated cases where contrails might even be cooling. We set the boundary between persistent contrails of any iRF and very strong ones at a value of iRF$\geq 19 \, \mathrm{W \, m^{-2}}$, which is at about the mean plus one standard deviation from the data of WGR21. For a contrail to become strongly warming, ice supersaturation must be high to allow large optical thicknesses and thus a large radiative effect.

For the case study, ERA-5 fraction of cloud cover, relative humidity, temperature, and wind components $u, v$, and $w$ from 21-03-2021 at 18 UTC were used. The data are given at pressure levels between 1000 hPa and 150 hPa. The spacing of the individual levels from 1000 hPa to 750 hPa is 25 hPa, from 750 hPa to 250 hPa is 50 hPa and from 250 hPa to 150 hPa is 25 hPa again. The area extends horizontally from 5°W to 21°E (geographic longitude) and from 48°N to 56°N (geographic latitude). However, the region of our interest is particularly from 5°W to 21°E at 52°N.

## 2.3 Forecast data

For the analysis of an example case (see section 3.2) ERA-5 and ICON-EU data were applied. The ICON model (ICOsahedral Nonhydrostatic model, Zängl et al., 2015), which was produced by the German Weather Service (DWD) and the Max-Planck Institute for Meteorology in Hamburg (MPI-M), is a global and regional numerical weather prediction model for short- and medium-range weather forecast. In the horizontal, the spacing of the grid, formed by projecting an icosahedron onto a sphere, is 13 km. In the vertical, the grid consists of 90 layers. The uppermost level is located at 75 km. Near the ground, the vertical layers follow the terrain. With increasing altitude, the signal of the terrain weakens, so that the layer height approaches a constant value. Forecast data is produced daily at 00, 06, 12, and 18 UTC. In addition, simulations with much more refined domains, so-called nests, are possible by bisecting the triangles of the original model. The ICON-EU, with a refined domain for Europe, is very closely linked to the global ICON forecast by two-way nesting. It has a horizontal grid spacing of 6.5 km. The vertical layers go up to level 60 and reach a maximum height of 22.5 km.

The cloud microphysics used in the ICON-EU model is the "two-category ice scheme" described in Doms et al. (2021). It is a single-moment scheme with prognostic variables for vapour, ice, and snow mass fractions. Cloud ice is assumed to nucleate at water saturation for $T > -25°C$, and below that temperature by heterogeneous deposition nucleation once ice saturation is surpassed or by freezing of supercooled droplets as soon as the temperature falls below $-37°C$. Ice crystals grow by vapour deposition with a rate proportional to the ice supersaturation. For computing radiation, the model includes a parameterisation of sub-grid scale cloudiness.

The data used for this analysis are obtained from the Pamore service of DWD (https://www.dwd.de/DE/leistungen/pamore/pamore.html ?nn=342666, accessed 18-01-2022). This server provides archived forecast data. Here we use the 6 hr forecast of ICON-EU data from a forecast run that started at 12 UTC on 21-03-2021. The data are retrieved on a regular $0.0625°.0625°$ longitude/latitude grid. We retrieved temperature, relative humidity, cloud coverage, and pressure on model levels 15-60 (about 150 hPa to ground).

## 2.4 Calculation of lapse rates

An important quantity to characterise the atmospheric stratification is the static stability which is $\gamma = -\mathrm{d}T/\mathrm{d}h$, i.e. the vertical temperature gradient (times minus one). Alternative expressions for the stability are the corresponding gradient of potential temperature and the Brunt-Väisälä frequency. Here we consider the lapse rate and the potential temperature gradient, but the results should be expressable in terms of the alternative as well. For orientation, we note that dry air in neutral stratification (where a dry air parcel lifted adiabatically does not experience any restoring force) has a dry-adiabatic lapse rate of $9.8\,\mathrm{K\,km^{-1}}$. The condensation (resublimation) of water vapour into ice reduces this value, but only slightly in the uppermost troposphere since the mass concentration of water vapour is very small at these levels. A mean lapse rate for the troposphere is about $6.5\,\mathrm{K\,km^{-1}}$. Further, we note that negative lapse rates are characteristic for the stratosphere and for temperature inversions in the troposphere.

From ten years of MOZAIC and ERA–5 data, a dataset was produced that was initially used to analyze the influence of natural weather variability on the instantaneous radiative effect of persistent contrails and in particular contrails with high iRF (see WGR21). The dataset is now the base for a statistical analysis regarding the effect of ISSRs and thin cirrus on the lapse rate. For this study, we use 403502 data records. Note that we take contrail persistence here as a proxy for ice supersaturation although the latter requires that also contrail formation is possible, which means essentially that the temperature must be lower than a threshold that is typically around $-40°$C. We found that the distributions presented below are hardly affected by this additional condition.

For every flight point, where MOZAIC diagnosed ice subsaturation (358141 cases), ice supersaturation (persistent contrails, 45361), and large $\mathrm{iRF} \geq 19\,\mathrm{W\,m^{-2}}$ (3961 cases), the lapse rate was calculated from ERA-5 temperature values on the neighbouring pressure levels and from these pressure values. It was derived from the barometric height formula, which relates a relative decrease of atmospheric pressure $p$ to an altitude increase:

$$\frac{\mathrm{d}p}{p} = -\frac{g}{R_d}\frac{\mathrm{d}h}{T(h)}. \tag{1}$$

Here $g$ is the gravitational acceleration (the slight height dependence will be neglected), $R_d$ is the special gas constant for dry air, $h$ is the altitude, and $T(h)$ is the temperature at altitude $h$. This equation can be integrated once $T(h)$ is given.

Dealing with model data or discrete profile data there is only information on $T$ at certain levels, here pressure levels, say $T_0$ at $p_0$ (lower level) and $T_1$ at $p_1$ (higher level). Without further information, it is justified to assume a constant temperature

gradient (or lapse rate) between these two levels. The height difference $\Delta h$ and thus the lapse rate $(T_0 - T_1)/\Delta h := \gamma$ can be calculated with this assumption. The integral of the barometric height formula is then:

$$\ln\left(\frac{p_0}{p_1}\right) = \frac{g}{R_d} \int_{h_0}^{h_1} \frac{\mathrm{d}h}{T_0 + \gamma(h - h_0)}. \tag{2}$$

Substituting $\vartheta$ for $T_0 + \gamma(h - h_0)$ leads to

$$\ln\left(\frac{p_0}{p_1}\right) = \frac{g}{\gamma R_d} \int_{T_0}^{T_1} \frac{\mathrm{d}\vartheta}{\vartheta} = \frac{g}{\gamma R_d} \ln\left(\frac{T_0}{T_1}\right). \tag{3}$$

Solving for $\gamma$ gives the desired result

$$\gamma = \frac{g}{R_d} \frac{\ln(T_0/T_1)}{\ln(p_0/p_1)}. \tag{4}$$

Thus, our procedure is as follows: Assume the MOZAIC aircraft flies on a flight level at a pressure $p$. Let $p_1 \leq p \leq p_0$, with $p_1$ and $p_0$ standard pressure levels of the ERA-5 data. Further, let $T_1$ and $T_0$ be the temperatures on these two levels, interpolated to the longitude-latitude and time of the MOZAIC record. Then an application of Eq. 4 yields the desired lapse rate. This procedure implies that a lapse rate could not be determined if the aircraft flew at $p \geq 300\,\mathrm{hPa}$ or at $p \leq 200\,\mathrm{hPa}$, since we did not retrieve data on pressure levels outside the range of 200 to 300 hPa.

For the statistical analysis of the lapse rate, cumulative distribution functions (cdfs) and probability density functions (pdfs) were produced for conditions of ice subsaturation, ice supersaturation, and strong iRF cases. For probability densities, the kernel density estimator was computed using the Epanechnikov kernel (Silverman, 1998). Five hundred bins were set for each of the pdfs of 2000 – 2009 with maximum and minimum following the lapse rate values. The optimal bin sizes were calculated by the program.

For completeness, we show how the calculation can be done in terms of potential temperature gradients. Evidently

$$\frac{\mathrm{d}\Theta}{\mathrm{d}h} = \frac{\mathrm{d}\Theta}{\mathrm{d}T} \frac{\mathrm{d}T}{\mathrm{d}h} = -\gamma \frac{\mathrm{d}\Theta}{\mathrm{d}T}. \tag{5}$$

In this equation we replace the derivative $\mathrm{d}\Theta/\mathrm{d}T$ by the difference quotient $(\Theta_1 - \Theta_0)/(T_1 - T_0)$, where the potential temperature is given by $\Theta_i = T_i(1000\,\mathrm{hPa}/p_i)^{(R_d/c_p)}$. The factor $\gamma$ is taken from Eq. 4.

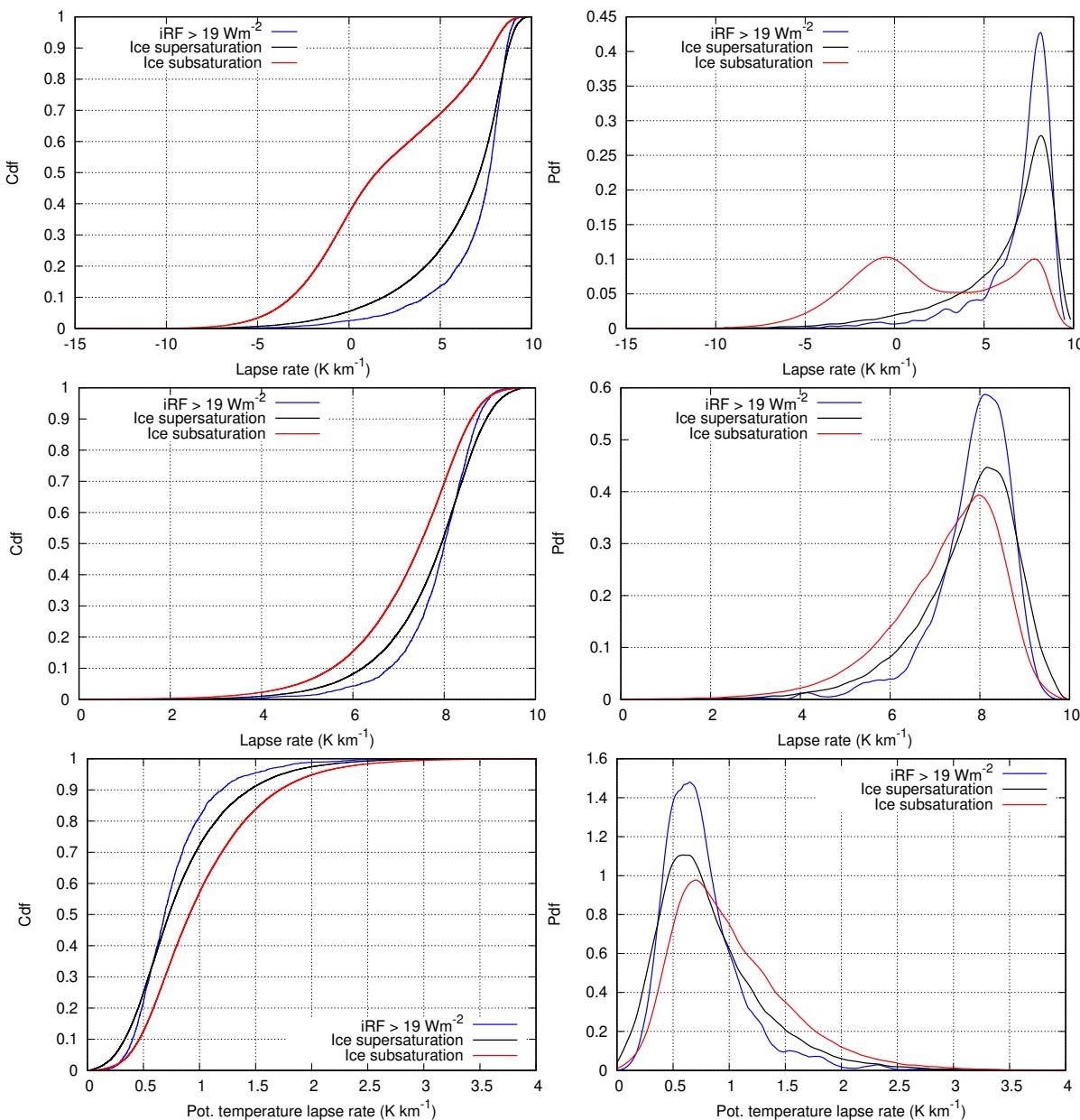

**Figure 1.** Top left panel: Cumulative distribution function of lapse rates, $-\mathrm{d}T/\mathrm{d}h$, for ice subsaturated (red) and ice supersaturated air masses. Top right panel: Probability densities for the same lapse rate distributions. The middle panels show the lapse rate distributions (left) and probability densities (right) excluding stratospheric cases, that is, the upper pressure level of each event has $PV \leq 2\,PVU$. Please note the different $\gamma$-ranges plotted in the upper and middle panels. The bottom panels show the corresponding troposphere-only distributions (left) and probability densities (right) for potential temperature gradients.

## 3 Results

### 3.1 Statistics over 10 years of data

Figure 1 shows conditional statistical distributions of $\gamma$ and $\mathrm{d}\Theta/\mathrm{d}h$ for locations and times with subsaturation (red curves), for ISSRs (black; strictly speaking, for regions that allow the formation of persistent contrails, see above), and for ISSRs that even allow the formation of contrails with strong iRF (blue). The upper left panel shows the three cumulative distribution functions for $\gamma$, and the upper right one the corresponding (and equivalent) probability density functions. The two middle panels show the distributions for cases that lay completely in the troposphere, i.e. the potential vorticity on the upper pressure level ($p_1$) does not exceed $2\,\mathrm{PVU}$. The two lowermost panels show the corresponding distributions and probability densities for the vertical gradient of the potential temperature, also for the troposphere-only cases. The representations in terms of $\gamma$ and $\mathrm{d}\Theta/\mathrm{d}h$ are equivalent. It is immediately evident that the red curves differ considerably from the other two. ISSRs are predominantly characterized by large lapse rates or small gradients of potential temperature, close to near neutral stratification. In the following, we concentrate on the tropospheric cases, since only in this case our assumption of a linear temperature profile is justified. If the tropopause is somewhere between the two involved pressure levels, the lapse rate at the location where the MOZAIC aircraft measured ice supersaturation, is underestimated. The stratospheric influence leads to many cases with low or negative lapse rates. Constraining the statistics to the troposphere lets many of the low values vanish and the pdfs concentrate more on the higher values. Indeed, the mean values for the whole data set (including the stratosphere) are $2.06, 6.11, 6.92\,\mathrm{K\,km^{-1}}$ for the non-ISSRs, the ISSRs and the ISSRs with high iRF, respectively, but these values increase to $7.23, 7.71, 7.85\,\mathrm{K\,km^{-1}}$ if only tropospheric cases are retained. Of course, the widths of the conditional distributions are considerably wider for the whole dataset than for the troposphere-only data. We note, that in all cases the mean and mode values of the distributions are larger for ISSRs (black and blue curves) than for non-ISSRs and the distributions are narrower for ISSRs and more concentrated to the peaks of the pdfs. For the tropospheric data, the peaks are at $8.00, 8.15, 8.10\,\mathrm{K\,km^{-1}}$, and the standard deviations are $1.29, 1.15, 0.89\,\mathrm{K\,km^{-1}}$ for non-ISSRs, the ISSRs and the ISSRs with high iRF, respectively. 70% of the dry cases have a lapse rate lower than $8\,\mathrm{K\,km^{-1}}$, but for the ISSR cases this fraction is lower, about 55%. These relations are more extreme if the stratosphere affects the analysis. Obviously, the transition from general ISSRs (persistent contrails) to the smaller class of ISSRs that allow contrails with large iRF and that require higher degrees of supersaturation leads to a narrowing of the lapse-rate distribution to the highest values (and to the lowest values of the distribution in terms of potential temperature), whereby the peak stays almost at the same value. That is, the mechanism that steepens temperature profiles gets stronger with increasing supersaturation. This will be discussed below.

### 3.2 An example case

In the following, an example case of a tall ISSR will be discussed in more detail, that occurred on March 21, 2021 at 18 UTC for several hours over Belgium and the Netherlands. For a geographic assignment and an overview of the synoptic conditions, Fig. 2 shows the geopotential height (in gpm) on the $250\,\mathrm{hPa}$ level and the locations of ice supersaturation (white contours, in %), obtained from ERA5.

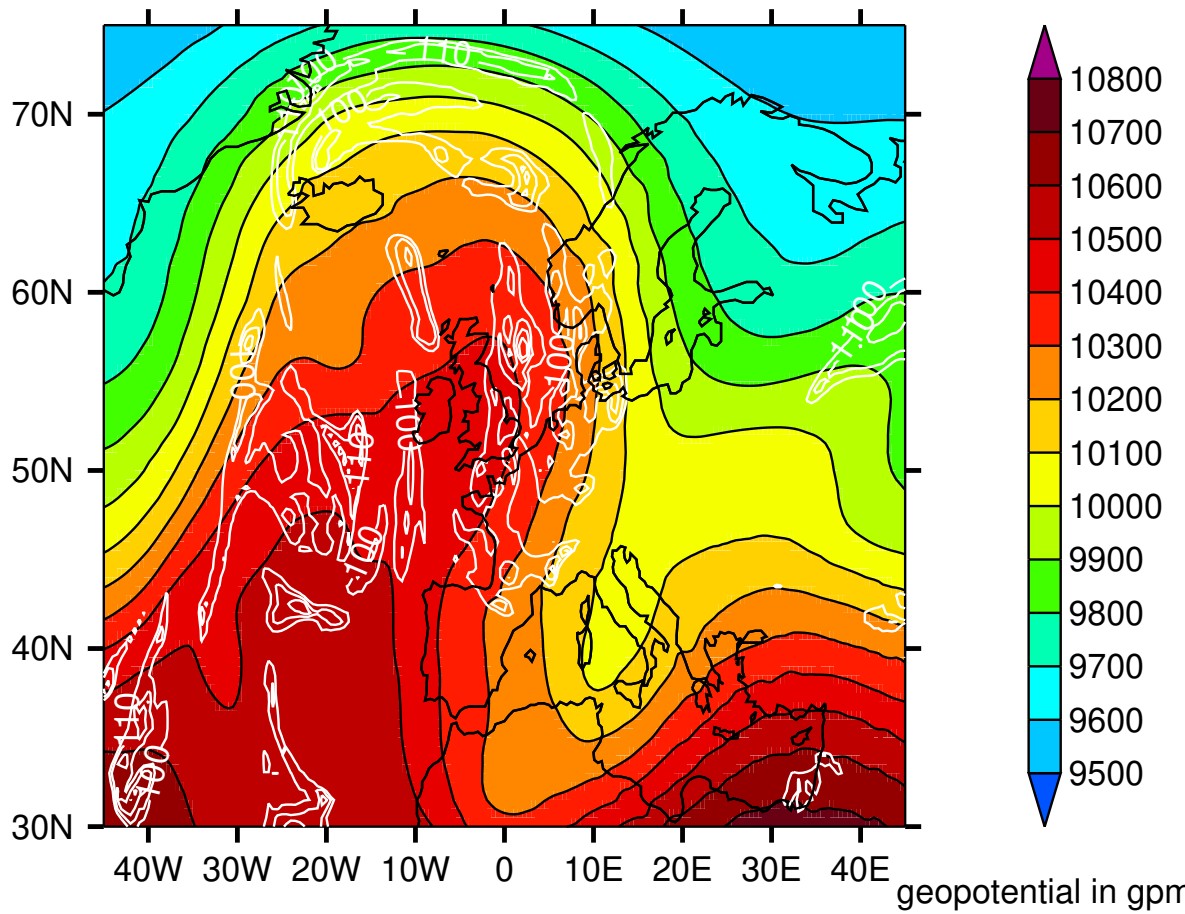

**Figure 2.** Geopotential height (colour coding) and ice supersaturation (white contours, starting at saturation, increment 10%) on the 250 hPa level at 18 UTC on 21 March 2021.

At the time of analysis, there is a high-pressure system located over the southern North Atlantic, which is connected to an anticyclonic flow, with a geopotential height of more than 10600 gpm. The resulting high-pressure ridge extends further to the north.

The area of Belgium and the Netherlands is influenced by upwards moving air masses, due to the location at the forefront of the high-pressure ridge. There, the geopotential is around 10300 gpm and high values of ice supersaturation are found (white lines). This large ISSR will now be examined in more detail.

In Fig. 3 (as in all further figures to come) we display a longitude-altitude cut through the ISSR along 52°N. The right panel of the figure shows how this case is simulated in the ERA-5 reanalysis where it is quite a large ISSR, that extends vertically from about 400 to 200 hPa, that is more than about 4 km, and zonally from about 0°E to 15°E, which is more than 1100 km. The relative humidity with respect to ice reaches and exceeds 130% at some locations. Moreover, the figure shows a

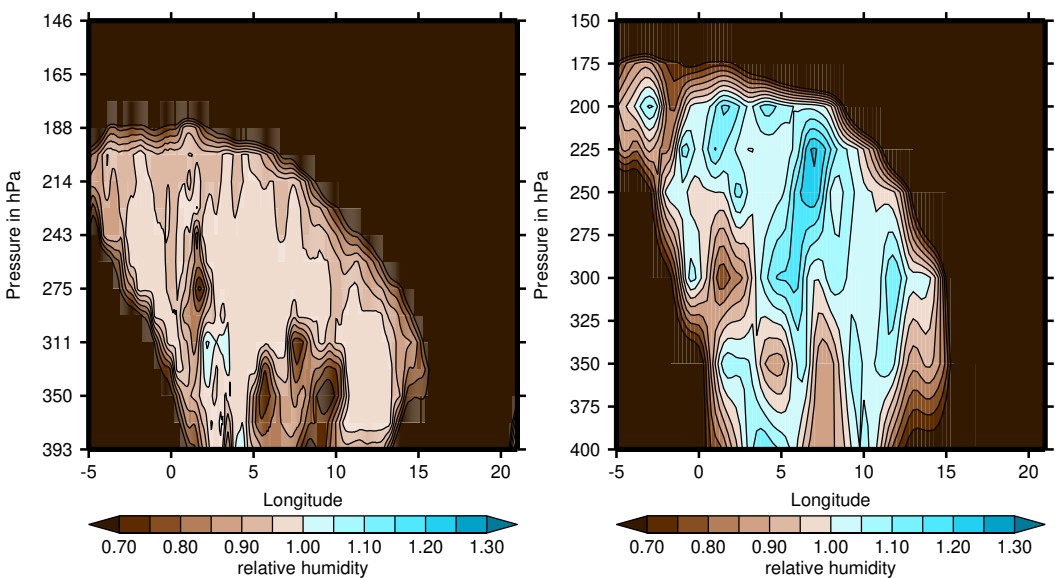

**Figure 3.** Latitude-altitude plots of relative humidity with respect to ice on 21 March 2021, 18 UTC, along 52°N. Left panel: data from ICON-EU. Right panel: data from ERA-5. In this figure, we display only the upper troposphere and lowermost stratosphere. Please note the different pressure scales in both panels.

remarkable small-scale structure with several locations of substantial subsaturation embedded in the supersaturated region. The

left panel reveals how this case appeared in the forecast of the ICON-EU model. Obviously, there is a similarly shaped region

of high humidity, but saturation or even supersaturation is reached only in small spots. (Probably the ICON microphysics only

achieves ice supersaturation in cases where liquid droplets, cooled down to their supercooling limit at −37°C, freeze. This

process starts at water saturation. It then depends on the phase relaxation time, how long ice supersaturation is maintained in

the model.) However, the relative humidity is above 95% over most of the locations where ERA-5 indicates supersaturation.

The embedded dry bubbles appear in both data sets at similar locations, with lower values in the ICON-EU data. The similarity

of these patterns is astonishing knowing the difficulties to predict the humidity fields on a small local scale. According to both

forecast and reanalysis, the ISSR or almost saturated region was connected to a cirrus cloud, as shown in Fig. 4. As clouds do

not form from subsaturated water vapour, this indicates that the ICON-EU model had supersaturation in this region not long

before. In the lower part of the cloud, the cloud fraction reaches 100% in the ICON-EU (less in the reanalysis), but the cloud

is quite thin with low cloud fractions in the upper and colder levels. The temperature field is shown for both models in Fig. 5.

They appear quite similar to each other. Stippling marks the location of the ISSR. The upper parts of the ISSR and the cloud

top have temperatures below 215 K. Such low temperatures imply slow ice growth processes, and as the cloud top is thin with

probably low crystal number densities, there are long relaxation times for the supersaturation (e.g. Khvorostyanov and Sassen,

1998). The models do not necessarily reflect this physics, because cloud physics parameterisations in weather- and climate

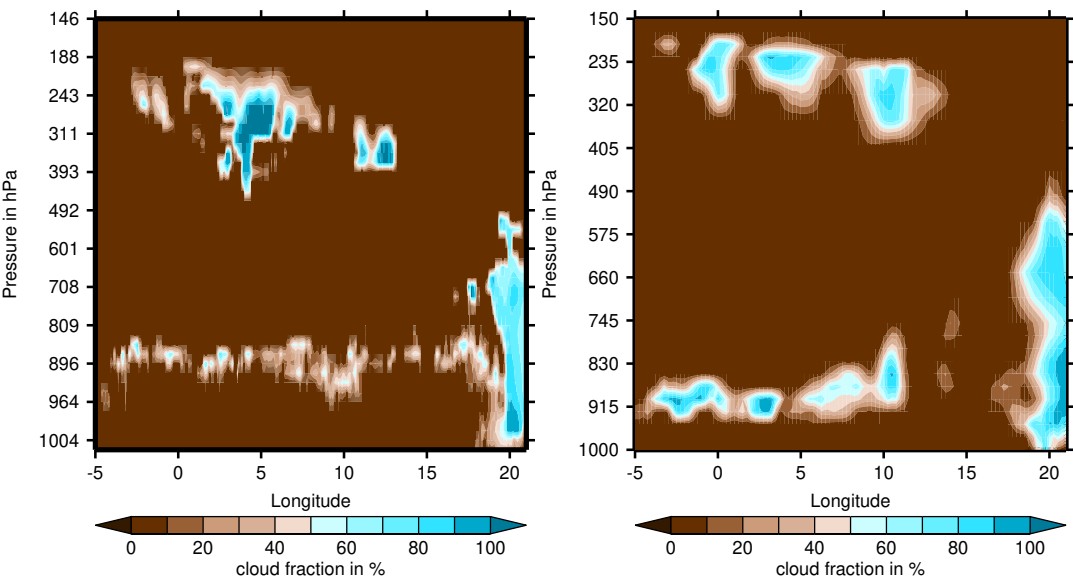

**Figure 4.** The cloudiness in the same situation, left: ICON-EU, right: ERA-5.

models often assume that the excess supersaturation within a cloud is converted to ice immediately (within the same or next time step); this is known as saturation adjustment.

The statistical results from above lead to the expectation that the lapse rate in the ISSR should be relatively large and this is indeed found in both data sets. Figure 6 shows the lapse rate field computed from the ICON-EU and the ERA5 data. Again the
235 patterns appear quite similar. At the position of the ISSR in the upper troposphere (marked by stippling) there is a maximum in the lapse rates with values exceeding $9\,\mathrm{K\,km^{-1}}$, that is, the stratification is almost neutral. Directly above the ISSR, at the lower boundary of the stratosphere, lapse rates are negative because temperatures increase upwards. Obviously, there is a quite shallow transition zone between almost neutral and very stable stratification (see also Birner et al., 2002; Birner, 2006). In addition, the strong reversal of the lapse rate at about $800\,\mathrm{hPa}$ is conspicuous: This is probably an inversion induced by
240 descending movements. Such inversions are typically caused by sinking air masses above a high-pressure system at ground level (subsidence inversion, see e.g., Malberg, 2002). There was indeed a high-pressure system centred southwest of Ireland and elongated in a southwest-northeastern direction with surface pressures exceeding $1020\,\mathrm{hPa}$ over Belgium and the Netherlands. Immler et al. (2008) found with a Lidar based at Lindenberg (southeast of Berlin, Germany) that ice supersaturation occurred quite frequently in anticyclonic systems and it was almost always found together with thin and sub-visible cirrus clouds.
Similarly, a climatological study using ERA-Interim showed that ISSRs occur more often in anticyclonic systems than in cyclones (Gierens and Brinkop, 2012). It is thus no surprise that the ISSR that we look at in this study is in an anticyclonic system as well.

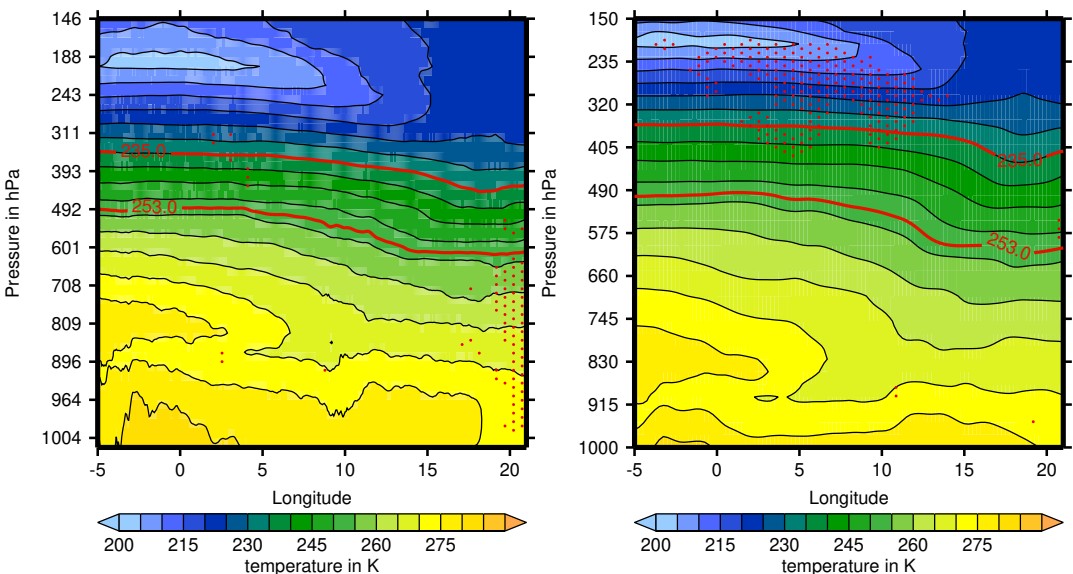

**Figure 5.** Left panel: Temperature field from ICON-EU. Right panel: Temperature field from ERA-5. The two red lines mark 253 K and 235 K for orientation. Below 253 K, the relative humidity in ERA-5 is defined with respect to ice; 235 K is the threshold for spontaneous freezing of water droplets, that is, below that temperature, condensed water is ice. Ice supersaturation is true at $T < 235$ K, while above that threshold it might be simply the vapour phase in a supercooled liquid cloud. Stippling indicates ice supersaturation, which at temperatures above 235 K implies the presence of a mixed-phase or supercooled-water cloud.

Figure 7 shows the field of potential temperature, $\Theta$, again for both models with similar patterns. The vertical gradient of $\Theta$ is quite small within the ISSR (which is equivalent to a near-neutral stratification), but above the ISSR it increases significantly upwards, indicating the transition to the stratosphere, i.e. the tropopause. The tropopause is higher above the ISSR than further to the east, and this longitudinal shape of the zone with the strong $\Theta$-gradient is characteristic for a weather situation with a western high and an eastern low at the ground, and the reverse situation in the upper troposphere.

The isohypses east of the ridge (see Fig. 2) show that the wind reaches 52°N approximately from the north, but with a western component. This is seen in Fig. 7 where the wind vectors within the ISSR show a westward direction. In the region with the ISSR, there is additionally upward motion due to convergence in the upper troposphere above the surface high pressure. This causes adiabatic cooling (perhaps there is also diabatic cooling due to the cloud), decreasing values of saturation pressures, and thus increasing relative humidity, which favours the formation of supersaturation. The uplifting air also shifts the tropopause upwards. All this results in a nicely coherent pattern with the ISSR directly below the tropopause. This is in general a preferred location for ice supersaturation (Spichtinger et al., 2003a; Petzold et al., 2020).

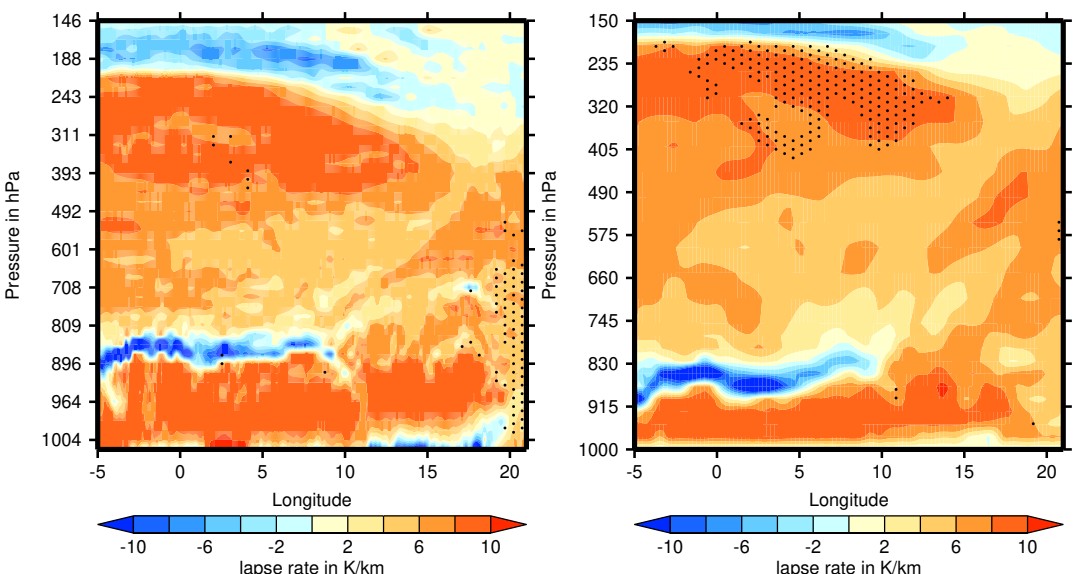

**Figure 6.** Longitude-altitude plot of atmospheric lapse rates along 52°N on March 21, 2021, at 18 UTC. Stippling indicates ice supersaturation, which at temperatures above 235 K implies the presence of a mixed-phase or supercooled-water cloud. Left panel: ICON-EU, right panel: ERA-5. The vertical axes indicate a mean pressure for each model level (right).

## 4 Discussion

While the statistical facts are clear, that ISSRs in the upper troposphere are regions with large lapse rates or near to neutral stratification, the physics behind this is not yet clear. But without a potential physical mechanism, the statistics could point out merely a correlation instead of a causal relation. In this section, we discuss, therefore, that indeed physical mechanisms exist that can explain high lapse rates within ISSRs. These are, on the one hand, the effect that lifting of an air mass has on its temperature profile, and, on the other, radiation.

### 4.1 Lifting

Ice supersaturation and cloud formation are generally the result of lifting air masses (Gierens et al., 2012). This leads to adiabatic cooling, thus decreasing saturation vapour pressure and increasing relative humidity until eventually ice supersaturation is reached which may be followed by the formation of ice crystals in the upper troposphere. Vertical air motion leads to changes in temperature profiles and thus induces changes in the lapse rate, $\gamma$. Subsidence (i.e. downward motion) lowers $\gamma$ and can lead to inversions (that is, $\gamma < 0$, as shown in our example). Lifting of air, on the contrary, generally leads to higher values of $\gamma$. A simple explanation for the observed differences in the distributions of $\gamma$ between ISSRs on the one side and subsaturated air masses on the other could thus be, that ISSRs are predominantly recently lifted air masses or that they are still rising, while subsaturated air would not rise any more or even be in a state of subsidence. This is certainly one part of the explanation, but it

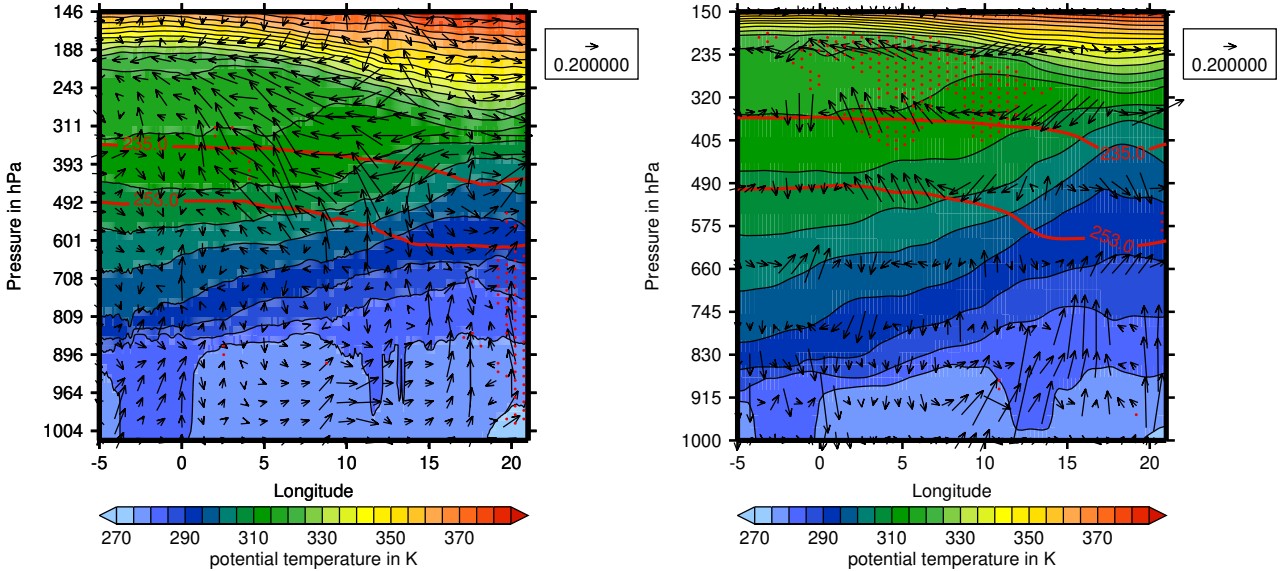

**Figure 7.** Left panel: Potential temperature field from ICON-EU. Right panel: The same field from ERA-5. The arrows show zonal and vertical wind components, differently normalised in both directions, so that the vertical motions are easily visible. Stippling indicates ice supersaturation, which at temperatures above 235 K implies the presence of a mixed-phase or supercooled-water cloud.

is not the whole story, because we must explain also why the lapse rates in ISSRs approach such high values, indeed almost the stability limit, the dry adiabatic lapse rate, $\Gamma = 9.76 \, \mathrm{K \, km^{-1}}$. Simple calculations show that it needs special initial conditions of lifting air masses, to achieve such high values of $\gamma$.

Lifting causes a steepening of temperature gradients as long as the relative humidity is below saturation in the whole layer that is lifted. When a layer is lifted, the pressure difference between its upper and lower boundary stays constant because of mass conservation. (We admit that there are no real boundaries in the air, but we "assume" them in the sense of a thought experiment). Thus, in case of adiabatic lifting, the upper boundary rises more than the lower boundary, with the consequence, that the temperature drop at the upper boundary exceeds that at the lower one. The same consequence holds for the water vapour saturation pressure; thus the relative humidity profile in the layer steepens (with the well-known consequence that cirrus clouds start to form at the top of moist rising air layers; see for instance simulation examples in Spichtinger and Gierens (2009), their figure 3, and the pattern of ISSR and cirrus evolution phases by Diao et al. (2013, 2015)). Thus it is quite typical that rising air masses reach saturation first at their top and later in the rest of the layer. As soon as saturation is achieved at the top, or rather as soon as condensation of water droplets or nucleation of ice crystals sets in, the release of latent heat slows down the steepening of the lapse rate, and $\gamma$ ceases to approach $\Gamma$; it approaches instead the smaller wet-adiabatic lapse rate. Admittedly, the effect of latent heat is weak in the relatively dry upper troposphere, but in simulations of lifting air masses of initially constant relative humidity, we were not able to achieve lapse rates exceeding about $8 \, \mathrm{K \, km^{-1}}$. The only possibility to achieve higher lapse rates

by lifting alone is a layer whose lower boundary is so much more humid than its top, that saturation is reached at the bottom before it is achieved at the top. This constitutes a conditionally unstable situation, where $\gamma$ can indeed rise to the stability limit, $\Gamma$.

Thus this discussion shows that lifting alone can lead to lapse rates considerably above $8\,\mathrm{K\,km^{-1}}$ only under the special initial condition of a sufficiently steep decreasing humidity gradient within the rising layer. It is improbable that only such special conditions lead to ice saturation after or during lifting. Therefore, we believe that lifting often brings $\gamma$ close to, say, $8\,\mathrm{K\,km^{-1}}$, but the observed highest values are caused by an additional diabatic process, that is radiation.

Additionally, all three conditional probability density functions of $\gamma$ peak at a value of about or slightly above $8\,\mathrm{K\,km^{-1}}$ (see Fig. 1). The peak height increases from subsaturated to supersaturated to strongly supersaturated, or from dry to moist and the distributions become narrower. This is surprising since we argued before that lifting does not lead to $\gamma$ much in excess of $8\,\mathrm{K\,km^{-1}}$, except in potentially unstable situations, and that in addition, the lapse rate tends to decrease as soon as condensation sets in. If this were all, the peaks of the lapse rate pdfs should appear at higher values for dry and lower values for supersaturated air masses. At least the peak height should decrease for increasingly moist air masses, with more of the distribution at lower lapse rates. This is, however, contrary to our results. Therefore, there must be a diabatic process that at least balances the $\gamma$-lowering effect of condensation and latent heat, and this effect is radiation.

## 4.2 Radiation

The explanation of the radiation effect needs three steps: 1) ISSRs mostly contain ice crystals (although they would not always be classified as a cloud), 2) Even without ice, the moist air has an effect on longwave radiation, with increasing strength with increasing humidity, and if ice crystals form the effect gets even stronger, 3) The radiative flux divergence leads to warming at the bottom of the ISSR and to cooling at the top, that is, the negative temperature gradient becomes more negative and the lapse rate increases.

When the modern research on ice supersaturated regions began more than 20 years ago, ISSRs were initially defined as supersaturated but cloud-free air masses. Later it became clear that this definition is too narrow; ice supersaturation does exist in cloud-free air, but it is much more connected to sub-visible cirrus (Gierens et al., 2000; Immler et al., 2008), and does also exist within (visible) cirrus clouds (Ovarlez et al., 2002; Spichtinger et al., 2004). Sun et al. (2011) find in a study combining satellite data from CERES (Clouds and the Earth's Radiant Energy System), MODIS (Moderate Resolution Imaging Spectroradiometer), and CALIPSO (Cloud-Aerosol Lidar and Infrared Pathfinder Satellite Observation) that about half of the CERES fields of view that MODIS would classify as clear (with a detection threshold for the optical thickness of about 0.3) are indeed covered with thin (subvisible) cirrus (as detected by CALIPSO, which has a much lower detection threshold than MODIS). At nighttime in particular, there is much subvisible cirrus, four times more than at daytime as the authors find. These thin clouds are mostly close to the tropopause (except in polar regions), and their water vapor mixing ratio is significantly larger than that of air masses where even CALIPSO does not detect any cloud, that is a purely clear sky, which is consistent with early results on ISSRs (Gierens et al., 1999; Spichtinger et al., 2003b). Sun et al. (2011) find also that when subvisible clouds

are present in the upper troposphere, they seem to affect the whole temperature and humidity profiles. Only the interaction of the subvisible cloud with radiation can probably cause such far-reaching influence.

Fusina et al. (2007) investigated the effect of enhanced humidity in ISSRs and of thin cirrus on radiation. They found that ice supersaturated layers are warmed from below by absorption of longwave radiation from the lower troposphere and ground and cooled at their tops due to the emission of radiation to space. It turned out that the warming vs. cooling difference increases with increasing relative humidity (at a fixed temperature, that is, the absolute humidity is the important factor). The effect on outgoing longwave radiation (OLR) is small, of the order $< 1\,\mathrm{W\,m^{-2}}$, but as soon ice is involved in the ISSR, the effect on OLR increases substantially, which is also consistent with the findings from Sun et al. (2011). Thus, radiation interaction with the enhanced humidity in ISSRs or with the ice crystals in thin (or subvisible) cirrus clouds can indeed balance the lowering tendency of $\gamma$. As stated above, lifting causes $\gamma$ to increase, but the rate of increase becomes slower the closer $\gamma$ is to the stability limit (except for potentially unstable cases, see above). Before condensation sets in, radiation might already act to let $\gamma$ rise quicker. Once condensation occurs, the latent heat effect of lowering $\gamma$ starts, but at the same time, the radiation effect gets stronger and may be able to at least balance the latent heat effect. The fact that the distribution is narrowest and most concentrated for ISSRs that allow strong contrails to form (blue curves), i.e. in the case of substantial supersaturation, indicates that radiation plays an additional role and confirms that it can indeed compensate for the $\gamma$-lowering tendency of condensation.

## 5  Conclusions

In the present study we have analysed the influence of ice supersaturation and thin cirrus cloudiness on the lapse rate within such regions. From a combination of ten years of measurement data from the MOZAIC/IAGOS project, from which we take the information on relative humidity at aircraft positions, with ERA-5 reanalysis data, from which we take the information on temperatures on adjacent pressure levels, we determine probability distribution and density functions, conditioned on three situations: 1) subsaturated airmass, 2) ice supersaturation (or allowing contrail persistence) and 3) situations with strongly warming contrails (instantaneous radiative forcing iRF$\geq 19\,\mathrm{W\,m^{-2}}$, a subset of 2). We find that the distribution functions for ISSRs and strongly warming contrails concentrate much to the high end of the lapse rate scale or, equivalently, to very small gradients of potential temperature, that is, to slightly stable to neutral stratification cases, in contrast to situations without or away from supersaturation, for which the lapse rates are less concentrated to high values. Lapse rates tend to be higher for ice supersaturated situations that allow high iRF than for the general ISSR, which points to a physical mechanism for steepening the temperature profiles whose efficiency increases with increasing supersaturation.

An example case was then investigated with a large ISSR below the tropopause in the ERA-5 data. For comparison, the same situation was studied as well with forecasts from the ICON-EU model of the German Weather Service. The higher-than-average lapse rate was indeed found in both the reanalysis and the forecast for the airmass that is ISS in ERA-5 and almost ISS in ICON-EU. Both data sets show cloudiness in the region of high relative humidity. The patterns of the humidity fields were remarkably similar, but ICON-EU is considerably drier than ERA-5. However, for all other variables we found a high degree

of agreement. In both cases, we find the enhanced humidity just below a tropopause that was pushed up by uplifting air, which obviously also led to increases in the relative humidity.

The major part of the statistical results can be explained by lifting. The common source of supersaturation and cloud formation, uplift, and adiabatic cooling of the air, cause the lapse rate in the uplifting layer of air to increase. Probably, the subsaturated layers have on average smaller lapse rates because a larger part of these layers is not in uplift or already in subsidence. But in ISSRs and in particular in regions with substantial supersaturation, the lapse rate should tend to lower values as soon as condensation or ice formation leads to a release of latent heat. This expectation is not corroborated by the position and width of the peaks of the probability density functions. Instead, the peaks for ISSRs are at higher values and narrower than the peak for subsaturated air. We deem that this can only be explained by the lapse-rate steepening effect of radiation which is stronger in ISSRs than in subsaturated air of the same temperature and which gets even more effective as soon as ice crystals form in the layer. That is radiation and latent heat push the lapse rate in opposite directions.

We can thus summarize this paper by stating the result that higher than usual lapse rates are another characteristic of ice supersaturated regions. This finding adds to the well-known ones, that ISSRs are on average colder and moister than their subsaturated environment.

*Author contributions.* KG, LW and SH conducted the research and wrote the paper, SR prepared the MOZAIC data and made them available.

*Competing interests.* The authors declare no competing interests.

*Acknowledgements.* The research described in this paper contributes to the DLR-internal project Eco2Fly which is coordinated by Volker Grewe. ERA-5 data are provided by the European Copernicus Data Service and ICON-EU data are provided via the Pamore system of the German Weather Service. The first author thanks Axel Kummerow (DWD) for his assistance in making these data available. The kernel density estimation was performed with an IDL program written by David G. Grier, Henrique Moyses, David Ruffner, and Chen Wang. The authors express their special thanks to Andreas Schäfler for his thorough reading and commenting a draft manuscript and to the two reviewers, Philipp Reutter and Ruben de Leon for reviewing the original version of the manuscript. Their questions and comments led to a considerable rethinking, new analyses, and major rewriting, that improved the paper and our interpretation of the results substantially. A discussion with Michael Ponater on the involved mechanisms is very much appreciated.

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
