# Peer review of "The effect of ice supersaturation and thin cirrus on lapse rates in the upper troposphere"

_Atmospheric Chemistry and Physics, 2022_

## Author Comment (AC1)

**Replies to the comments of the two reviewers**

**General remarks**:**

The two reviews by Philipp Reutter and Ruben Rodriguez Deleon were extremely helpful to not only improve the paper but also to clarify issues that we have not seen so clearly when we submitted the original version of the paper. You will see this by the major changes that we made. Most important in this respect was the hint that our analysis may suffer from having a stratospheric influence on the data. Indeed, this was the case. We corrected this and this also led us to see that the role of radiation is more modest than we thought initially.

In the following, we repeat the reviewer's comments in Calibri font. Our replies begin with the word "Reply" and are printed in Courier font.

The new version of the manuscript is produced with new text in red, and deleted text is marked as deleted. (Minor changes in grammar and spelling are generally not marked).

**Comments by Philipp Reutter and replies**

In this study, the authors deal with the lapse rate in differently humid air masses (subsaturated, supersaturated, big hit) along the flight paths of passenger aircraft. IAGOS measurement data as well as reanalysis data (ERA5) and forecast data (ICON-EU) are used. It is shown that in humid air masses the temperature decrease with altitude is stronger than for dry air masses.

Overall this manuscript present very interesting results. It fits very well within the scope of ACP. However, some points have not become entirely clear to me. In addition, some aspects should be formulated more precisely. Therefore, I would like to recommend this manuscript for publication with major corrections.

**Specific questions and comments:**

For me it is not clear, why you need the IAGOS data set for this investigation. I understand that comparing measurements with model data is helpful. However, such a statistic could also have been produced without the flight paths from IAGOS. Perhaps you could motivate the use of the IAGOS data more precisely in the text. How is the IAGOS data connected to the case study? Are there measurements available for this event?

Reply: Indeed, the justification for use of IAGOS data is missing in the paper. But it is simple: we use IAGOS RHi to determine whether there is actually ice supersaturation at a certain point and time. We use this only for the statistical part of the analysis. In Gierens et al. 2020 (Gierens, Matthes, Rohs, 2020) we showed that ERA-5 differs quite often from MOZAIC measurements in terms of RHi. Therefore, we wanted to use in-situ "truth" for the statistical investigation. To clarify this, we add a sentence in Sect. 2.1 "Data from commercial aircraft".

The exact calculation of the lapse rate, which is included in the statistics, could be explained in more detail. Is the lapse rate calculated from three model levels ("lapse rate was calculated from ERA-5 temperature values on the neighboring pressure levels and from these pressure values" P5L121) or

only from two levels ("...without further information, it is justified to assume a constant temperature gradient (or lapse rate) between these two levels.", P5L128)?

Reply: Sorry that this is not clear enough. We use the temperatures on the two pressure levels surrounding the point where the IAGOS aircraft made its measurement. That is, with ERA-5 pressure levels of 300, 250, 225, and 200 hPa, when IAGOS flew on, say, 270 hPa, we use temperatures on 300 and 250 hPa to compute  $\gamma$ , with the formula given in the paper. This implies also that  $\gamma$  could not be determined for flights at 300 hPa or a lower altitude, neither at 200 hPa nor a higher altitude.

We add a short paragraph after Eq. 4 that illustrates the procedure.

In my view, using the lapse rate it is not intuitive to see what is meant by a "large lapse rate". I would rather use "unstable", "neutral", "stable" for the lapse rate. This brings me to the next point

Maybe it is more convenient to use the gradient of the potential temperature, which you also mention in the beginning of 2.4 as a different expression for the stability. This makes it easier to interpret the results, from my perspective. If you want to stay with the temperature gradient, can you motivate this in more detail?

This can easily be achieved, as we have T and p on both levels we also have the potential temperature,  $\theta$ . We have  $\Delta h$ , so we can compute  $\Delta \theta / \Delta h$ .

In the paper, we provide now at the end of section 2.4 the formula to compute the lapse rate in terms of potential temperature.

New panels with the corresponding distributions of  $\Delta\theta/\Delta h$  are added to Figure 1 and describing text is added in 3.1.

By using IAGOS flight paths you most likely have a significant amount of data points within the lowermost stratosphere. How does this influence your results? How many of your data points are above the tropopause? Could there be a contribution to the pdf peak around 0 K/km for the ice subsaturation?

Reply: You touch here something very important that we admit we gave too little attention to. But you are absolutely right. If the tropopause is between the two pressure levels that are used to compute the lapse rate, the assumption of a linear temperature profile is violated with the consequence that the resulting lapse rate underestimates the true one. We have now corrected this flaw and made a new analysis where we only retained data where the upper pressure level is below the tropopause (it has  $PV \le 2PVU$ ). This avoids the mentioned underestimation and the effect can be seen in the comparison of the upper (original) and middle panels (only troposphere) in the new Fig. 1. All distributions shift to higher values of  $\gamma$ . But still, the values for ISSRs are more concentrated to high values than the values for dry cases. The fact that the pdfs for ISSR peak at higher  $\gamma$  and are narrower distributed than the pdf for the dry cases cannot be explained by lifting alone, since condensation in some ISSRs would cause a lowering trend of  $\gamma$ . Thus, there must be a process that inhibits this lowering. Although the

radiative effects are weak and slow, they seem to suffice to keep the lapse rates in ISSRs at their observed high values.

The text of section 3.1 has been changed considerably and new panels have been added to Fig. 1.

Since "big hit" is a rather new term, a brief explanation would be helpful for the reader. In Wilhelm et al. (2021) this term does not occur. How were "big hit" cases defined in this study? This simplifies the reproducibility of the study.

Reply: Meanwhile we think that this "working notion" is a bit misleading and not appropriate for our data. A "big hit" is actually meant to be a contrail with a substantially higher than average warming effect on climate. But this must refer to a contrail as a whole over its complete lifetime. However, from IAGOS we can only compute the instantaneous radiative forcing value of contrails at the very moment and location of their formation. As there is not necessarily a positive correlation between these instantaneous values and the contrail warming effect over its lifetime, we decided to avoid the notion "big hit" for the current paper. We will instead speak of contrails with very high iRF>19 W/m2, which is roughly the iRF mean plus one standard deviation.

Text and Figures will be adapted accordingly.

Regarding the contour figures: please, use the same vertical axis for plotting the ICON-EU and ERA5 results. It is very hard to compare the images side by side with slightly shifted axes. For some figures you show only values above approx. 400 hPa, for other figures you show the whole troposphere. Maybe focus on 500 hPa upwards?

Reply: We agree that it is not optimal to have different scales on the two y-axes. We looked into our manuals whether there is a simple way to make them equal. Unfortunately, it seems that we must interpolate the Icon data to the ERA-5 pressure levels for such a purpose. As interpolation introduces uncertainties, we refrain from following your advice.

Section 4.2: This section seems somewhat speculative to me.

When there is an additional contribution to the lapse rate increase by radiation, why do have all curves their (local) maximum at around the same lapse rate in Fig. 1? Isn't that contradicting the statement, that the second maximum of the sub saturated pdf is due to the fact, that the air is just too dry? So my question is, why the maxima of all curves overlap. Shouldn't the "big hit" curve then have the highest lapse rate because the radiative effect is strongest, followed by the supersaturated and then subsaturated curve?

Reply: This is a good question. We think we have overemphasised the radiation effect in the original manuscript. But still, radiation has an important role to explain the statistical findings. We give now new arguments in the section on lifting. There we explain that contrary to what we get, we would expect that the pdfs for moist cases peak at *lower* values than the pdf for dry cases. This is, because in moist cases, as soon as there is condensation, latent heating lowers the lapse rates in the direction of the wet-adiabatic value that is relevant for the given situation. There is no such

lowering for lifting dry layers. In the latter the lapse rate approaches the stability limit asymptotically, that is the rate of approach gets slower and slower the closer  $\gamma$  gets to  $\Gamma$  (9.76 K/km). In fact, values exceeding 8 K/km are hard to reach, except in potentially unstable cases where the relative humidity must decrease sufficiently through from bottom to top, so that condensation starts at the bottom of the layer while no condensation happens at the top. This is however a special, not a general or typical, situation. We cannot assume that ISSRs more often start as potentially unstable layers than drier layers. Thus, the only explanation for the fact that the peaks of the ISSR and iRF>19 W/m2 pdfs are at higher, not lower, values of  $\gamma$  than the peak for the subsaturated cases, is that there must be another process that keeps  $\gamma$  high despite the lowering tendency caused by latent heat. This must be radiation.

These considerations made a completely new version of the discussion section "lifting" necessary. Of course, radiation plays a more moderate role than we thought initially, but still, it plays a role and we need it for a complete explanation of the statistical facts.

Perhaps a little more light could be shed on the matter by carrying out simple radiation calculations for one exemplary profile per class (subsaturated, supersaturated, big hit), see e.g Fig. 11 of Fusina & Spichtinger, 2010. (Fusina, F. and Spichtinger, P.: Cirrus clouds triggered by radiation, a multiscale phenomenon, Atmos. Chem. Phys., 10, 5179–5190, https://doi.org/10.5194/acp-10-5179-2010, 2010.)

Reply: As stated above, we seek for a process that balances the  $\gamma$ lowering effect of condensation, and we think that only radiation can be this process. A numerical experiment to demonstrate this would therefore require both, lifting and radiation (and of course microphysics of condensation or ice formation). We think such simulations would need a lot of work, and the description of the model, the setup, the simulations, the interpretation of the results etc. would be enough material for a Master thesis and a new paper. This is far beyond the current investigations.

We have also rewritten the final part of the radiation subsection. As stated, we feel that we initially overstated this effect, but the new text should better fit the more modest role of radiation.

**Minor:**

**Please indicate, which ice microphysics is used in ICON-EU**

Reply: This is the "two-category ice scheme" (cloud ice and snow). It is described in "A Description of the Nonhydrostatic Regional COSMO-Model, Part II Physical Parameterizations" by Doms et al., 2021, doi: 10.5676/DWD pub/nwv/cosmo-doc\_6.00\_II.

New text has been added in sect. 2.3 to describe this.

P7L168: "quite a large ISSR" - maybe a bit more precise, since, as you mentioned, there are also patches of subsaturated air.

Reply: Yes, we agree. In the figure we see ice supersaturation at least from 400 to 200 hPa, which in the standard atmosphere extends from about 7100 m to 11800 m, that is, this region is more than 4 km high. Further, the region extends from about 0°E to 15°E along 52°N, which is more than 1100 km. These dimensions are quite unusual for an ISSR if you compare these numbers with those given in some older papers (Gierens and Spichtinger, 2000 and Spichtinger et al. 2003a; both are given in the list of references.)

We add more text in the description of this case.

**P8L171 Please discuss here briefly, why ICON-EU shows lower saturation values than ERA5. What is the difference between the two models which could explain this behavior?**

Reply: This is a good question. To answer it truly, we would have to run both models in parallel. As we don't have access to the models we must resort to some speculation. In the new text that describes the ICON model we write that at T<-25°C, heterogeneous ice nucleation can form ice and below -37°C water droplets will freeze spontaneously. Heterogeneous nucleation in the model seems to start at ice saturation, leaving not much space for supersaturation. Freezing of droplets, of course, starts at water saturation, since there are liquid droplets. This can provide some ice supersaturation. Perhaps the time scale for the complete consumption of the water vapour in excess of saturation is short (or assumed to be done within one time step as in the IFS). This again would constrain the possibility to maintain a large RHi.

Since this is speculation we dare only to put a sentence in brackets:

"(Probably the ICON microphysics only achieves ice supersaturation in cases where liquid droplets, cooled down to their supercooling limit at \$-37\$\Celsius, freeze. This process starts at water saturation. It then depends on the phase relaxation time, how long ice supersaturation is maintained in the model.)"

**P9L192 The feature in 800 hPa is not of interest for this topic, maybe show results only above 500-400 hPa (see my point 7)**

Reply: Yes, for this topic it is not directly relevant, we agree. But, it might fit into a larger picture that connects features of the general synoptic situation (with the fronts and corresponding air motions) with the situation in the upper troposphere. This is of course far beyond the current topic, but maybe interesting for somebody who wants to see the complete picture. Therefore, we prefer to leave this feature in the paper.

**Is Fig 5 necessary? Maybe include certain temperature isolines into Fig. 3 or 4?**

Reply: As this is a matter of taste and as figure 5 does not in any way mislead the reader, we think we can leave it as it is.

Figure 5/6/7: why so many red points (ice supersaturation) in lower regions for ICON-EU compared to ERA5? Additionally, the stippling is not mentioned in the figure caption of Fig 5 and Fig 7.

Reply: Sorry that we forgot to describe this. It is simple: Wherever there are water or mixed-phase clouds in the data, there is ice supersaturation and the plotting program then puts stipples there. That is, the stippling in the layers warmer than 235 K simply marks clouds.

We add now to the figure caption of fig. 5: "Stippling indicates ice-supersaturation, which at temperatures above 235\,K implies the presence of a mixed-phase or supercooled-water cloud."

Similar explanations are given now in figs. 6 and 7.

**Typos**

**P1L24 "and and how long..."**

Done

**P2L33 line break**

We include suggestions for hyphenation in the latex file, but once this text is printed in two-column mode, it will look different anyway.

**P2L42 remove "even" ?!**

Done

**P4L98 line break**

This is again something that will appear differently in the twocolumn setting of ACP. As this is a long link, hyphenation is inappropriate and so we suggest waiting for the ideas of the Editorial Office's typeset-professionals.

**Comments by Ruben Rodriguez Deleon and replies**

The study analyses the correlation between larger atmospheric lapse rates and the presence of supersaturation and thin cirrus clouds. It proposes that uplift and adiabatic cooling of an air mass would not result in such high observed lapse rates and suggests that atmospheric radiation effects must be a contributing factor, further enhanced by the presence of ice crystals, given their larger interaction with radiation. One strength of the study's approach is the combination of in situ and reanalysis and model data, although it is not clear from the manuscript how were the "big hit" cases defined and determined. I would recommend the article's publication after this aspect is addressed.

Reply: Meanwhile we think that this "working notion" is a bit misleading and not appropriate for our data. A "big hit" is actually meant to be a contrail with a substantially higher than average warming effect on climate. But this must refer to a contrail as a whole over its complete lifetime. However, from IAGOS we can only compute the instantaneous radiative forcing value of contrails at the very moment and location of their production. As there is not necessarily a positive correlation between these instantaneous values and the contrail warming effect over its lifetime, we decided to avoid the notion "big hit" for the current paper. We will instead speak of contrails with very high iRF>19  $W/m^2$ , which is roughly the iRF mean plus one standard deviation.

Text and Figures will be adapted accordingly.

Some other suggestions may help the reader understand the methodology and the interpretation of the results:

It is clearly stated in the manuscript that the highest lapse rates were found in Big Hit regions, but it is not clear how are these regions identified from the IAGOS data. In other words, is the presence of ice crystals identified and if so, how are natural cirrus and contrails discriminated? This would also help clarifying the sentence in In 8, which mentions the interaction between high ISS in Big Hit regions and radiation only, leaving out the interaction with ice crystals.

Reply: Indeed, the explanation of "Big Hit" and how we define and compute it was missing in the manuscript. We are sorry for that. In the revised version we include a new paragraph in section 2.2 that explains that we compute an instantaneous radiative forcing for each MOZAIC position where a persistent contrail is possible acc. to MOZAIC T and RHi. We define Big Hits as cases where the iRF exceeds its ten years average plus one standard deviation, but we now avoid the misleading notion "Big Hit" and write instead "strong iRF cases" or similar expressions. The details of the calculation are not given, as they can be found in the quoted papers by Wilhelm et al., 2021 and Schumann et al. (2012).

Following up on the previous comment, it would be helpful to explain how are the blue and the black lines in Fig. 1 determined in practice from the IAGOS database. From the manuscript it is not possible to understand how this was done. And one of the main outcomes of the study relies on this discrimination.

Reply: Yes, this was omitted, too, but now it can be found in the new paragraph in section 2.2, see Reply to point 1. The distinction is quite simple: The black curves represent all MOZAIC data records where a persistent contrail was possible (i.e., Schmidt-Appleman criterion fulfilled AND RHi $\geq$ 100%), the blue ones a subgroup of them, namely those that have an instantaneous radiative forcing iRF $\geq$ 19 W/m2.

It is not clear from the manuscript why without radiative interaction the results cannot have a physical explanation. A back of the envelope calculation to exemplify this would help the reader understand the need of the radiation hypothesis.

Reply: We repeat here the answer that we have given to a related question by Ph. Reutter.

We think we have overemphasised the radiation effect in the original manuscript. But still, radiation has an important role to explain the statistical findings. We give now new arguments in the section on lifting. We explain there that contrary to what we get, we would expect that the pdfs for moist cases peak at lower values than the pdf for dry cases. This is, because in moist cases, as soon as there

is condensation, latent heating lowers the lapse rates in the direction of the wet-adiabatic value that is relevant for the given situation. There is no such lowering for lifting dry layers. In the latter the lapse rate approaches the stability limit asymptotically, so that the rate of approach gets slower and slower the closer  $\gamma$  gets to  $\Gamma$  (9.76 K/km). In fact, values exceeding 8 K/km are hard to reach, except in potentially instable cases where the relative humidity must decrease sufficiently through from bottom to top, so that condensation starts at the bottom of the layer while no condensation happens at the top. This is however a special, not a general or typical, situation. We cannot assume that ISSRs more often start as potentially instable layers than drier layers. Thus, the only explanation for the fact that the peaks of the ISSR and iRF>19 W/m2 pdfs are at higher, not lower, values of  $\gamma$  than the peak for the subsaturated cases, is that there must be another process that keeps  $\gamma$  high in spite of the lowering tendency caused by latent heat. This must be radiation.

These considerations made a completely new version of the discussion section "lifting" necessary. Of course, radiation plays a more moderate role than we thought initially, but still it plays a role and we need it for a complete explanation of the statistical facts.

**Minor comments: ALL DONE**

**Ln 24, repeated "and".**

Done

**Ln 154 delete "the"**

Done

**7 legend, delete "zonal"**

Done

**Ln 291 "distribution" instead of "distributions"**

Done